# REWARD AS OBSERVATION: LEARNING REWARD-BASED POLICIES FOR RAPID ADAPTATION

## ABSTRACT

This paper explores a reward-based policy to achieve zero-shot transfer between source and target environments with completely different observation spaces. While humans can demonstrate impressive adaptation capabilities, deep neural network policies often struggle to adapt to a new environment and require a considerable amount of samples for successful transfer. Instead, we propose a novel reward-based policy only conditioned on rewards and actions, enabling zero-shot adaptation to new environments with completely different observations. We discuss the challenges and feasibility of a reward-based policy and then propose a practical algorithm for training. We demonstrate that a reward policy can be trained within three different environments, Pointmass, Cartpole, and 2D Car Racing, and transferred to completely different observations, such as different color palettes or 3D rendering, in a zero-shot manner. We also demonstrate that a reward-based policy can further guide the training of an observation-based policy in the target environment.

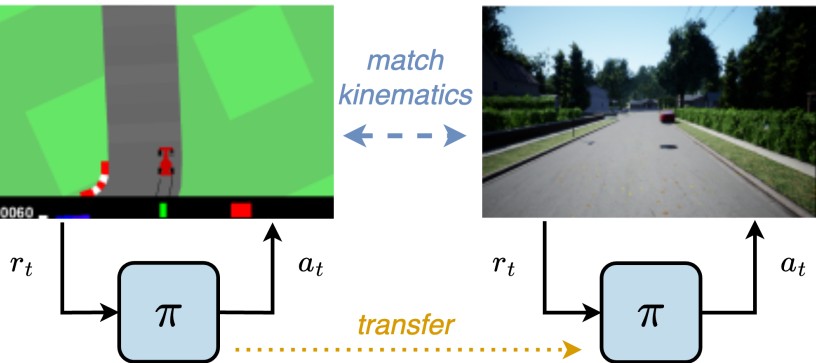

Figure 1: A reward-based policy enables zero-shot transfer from the 2D Car Racing environment to the 3D AirSim environment by taking reward and action histories as the only input. We assume that kinematics between environments are matched.

## 1 INTRODUCTION

Humans are known for their remarkable ability to adapt to completely new environments. For example, once gamers master a racing game, they can adapt rapidly to new games with entirely different visuals and themes. People can also navigate their surroundings in completely new cities despite unfamiliar environments. Therefore, researchers have investigated the development of intelligent agents that are capable of rapid adaptation in unseen environments. However, neural network agents trained with deep reinforcement learning (deep RL) typically struggle in unfamiliar environments, even when trained with massive data. Therefore, developing robust agents that perform well in unseen environments remains an open challenge in various fields, such as machine learning, artificial intelligence, and robotics.

This paper discusses the problem of rapid adaptation from a source problem to a target problem, where significant changes occur only in observation spaces. This change in observation reflects

various adaptation scenarios, such as altering color palettes, changing screen resolutions, switching rendering engines from OpenGL to Unreal Engine, or transitioning from 2D to 3D visualizations, all without making assumptions. We also assume that ground-truth state information is not accessible to the policy in the target environment, following the definition of a Partially Observable Markov Decision Process (POMDP). This assumption reflects situations where the internal programming variables of a game or the ground-truth hardware states of a robot are inaccessible to the control policy.

Our key hypothesis is that a reward can serve as a strong signal for rapid adaptation if it is sufficiently dense. This work explores the development of a reward-based policy that takes only rewards as input, disregarding observations. In the context of computer games, this would be analogous to playing a game solely by observing the score. If successful, our reward-based agent could be deployed across different environments with any visual renderings in a zero-shot manner. We note that assuming the availability of rewards to the agent is somewhat unconventional. However, this assumption is not entirely unreasonable, as POMDPs provide per-step rewards to deep RL algorithms anyway. Furthermore, we will later relax this assumption by demonstrating that rewards can be more easily estimated from observations than high-dimensional state vectors.

In this work, we aim to explore the novel concept of a reward-based policy for rapid transfer, which is only conditioned on the history of rewards and actions without observations. First, we discuss our problem assumptions, such as significant changes in observation spaces while other MDP components remain the same. Then, we introduce a reward-based policy, followed by discussions on its difficulties, feasibility, requirements, and sub-optimal behaviors. Finally, we propose a practical framework for training a reward-based policy, which is enabled by two key components: (1) a temporal history of the reward/action pairs and (2) the guidance of an expert policy.

We demonstrate that the proposed framework can successfully train a reward-based policy in three environments: Pointmass, Cartpole, and Car Racing. The learned reward-based policies exhibit reasonable behavior in all three environments, achieving approximately 60% to 90% of the performance of standard observation-based policies. However, due to their independence from observations, they can be transferred to novel environments with significant observation shifts in a zero-shot manner, including palette swaps or even 2D-to-3D transfers. If the reward function is unavailable, we can rapidly learn a reward estimator for rapid transfer, which performs significantly better than training a state estimator. Lastly, we will conduct an ablation study on the effect of expert guidance.

## 2 RELATED WORK

Transfer learning in RL leverages existing information for rapid adaptation in a new scenario, which has been approached by a wide range of algorithms. One possible approach is learning from demonstration (Schaal, 1996), where expert demonstrations are available to guide the learning process. Offline methods can use these demonstrations for offline RL (Ma et al., 2019), (Yang & Nachum, 2021), (Li et al., 2023) and for pretraining to learn value functions or policies (Silver et al., 2016), (Stachowicz et al., 2023). Online methods use expert demonstrations to improve policy exploration during learning, and include policy iterations (Piot et al., 2014), (Chemali & Lazaric, 2015), policy gradients (Nair et al., 2018), (Kang et al., 2018), and Q-learning approaches (Brys et al., 2015), (Hester et al., 2018).

Another way to transfer knowledge is representation learning, which establishes reusable representations in the action or state spaces that can be shared between source and target environments. Progressive networks (Rusu et al., 2016) and PathNet (Fernando et al., 2017) directly train common reusable representations and leverage them during policy training. There exists a line of works that split networks into general and reusable modules (Andreas et al., 2017), (Devin et al., 2017). Gupta et al. (2017) learns an invariant feature space in order to better transfer between tasks. Bou-Ammar & Taylor (2011), Taylor et al. (2007) learn inter-task mappings to deal with state differences between tasks. Successor representations (Dayan, 1993) seek to separate the environment dynamics from the reward function. These have seen widespread use (Barreto et al., 2018), (Zhang et al., 2017), (Konidaris & Barto, 2006) for transferring knowledge for different tasks with the same state and action spaces. Zhang et al. (2018) does a similar approach of separating dynamics and reward for learning a better policy for transfer. We avoid the need for dealing with different observations between the source and target environments by using the shared reward function as the policy input.

The most relevant method of transfer for our work is to transfer at the policy level. A popular method to transfer involves distilling information from an expert policy. This method uses supervised learning to match action distributions between the student and teacher policies, and is widely used for robotic applications (Kumar et al., 2021), (Liang et al., 2023), (Miki et al., 2022). Policies can also be directly reused for updating policies in a transfer setting, as in (Barreto et al., 2017), (Fernández & Veloso, 2006), (Rajendran et al., 2017). This is the closest domain to our method, which directly transfers an expert policy via reward information.

Another very similar sounding work to ours is the reward-conditioned policy (Kumar et al., 2019). However, this work trains standard observation-based policies with an additional reward conditioning that allows for using suboptimal trajectories as optimal supervision. This is in contrast to our work which uses no observation feedback and uses the reward directly as the policy input.

## 3 REWARD-BASED POLICY

### 3.1 PROBLEM DEFINITION

A Partially Observable Markov decision process (POMDP) is a popular tool for modeling a sequential decision problem. It is defined as a tuple $(S, O, A, R, T)$, where $S$ is the state space, $O$ is the observation space, $A$ is the action space, $T$ is the transition function that defines how states change given an action, and $R$ is the reward function. The policy $\pi : O \mapsto A$ takes action based on the given observation without knowing the underlying state $s \in S$, and we want to find an optimal policy that maximizes the discounted cumulative sum of the reward over time: $\sum_{t=0}^{T} \gamma^t R(s_t, a_t)$.

Our scenario defines the transfer problem as training a reusable policy from a source to a target environment. We assume the source and target environments should have the same transition, reward, action, and state but may have very different observation spaces. Therefore, when the source environment is described as a tuple $(S, O, A, R, T)$, the target environment becomes $(S, \tilde{O}, A, R, T)$, where there is no assumption between $O$ and $\tilde{O}$. For all environments we consider, we assume the availability of a dense reward, which makes the training of a reward-based policy feasible.

### 3.2 REWARD-BASED POLICY: DEFINITION AND PROPERTIES

Our paper investigates a novel reward-based policy $\pi : R \times A \mapsto A$, which takes only the previous rewards and actions as input. Our key intuition is that a typical observation-based policy for transfer inevitably requires an additional training process to map the observations from the target environment to match those of the source environment, which requires considerable training samples. However, a reward-based policy can immediately be deployed in a zero-shot fashion if the underlying system dynamics remain the same.

**Difficulties.** Learning a policy from only reward information is more difficult than using a standard observation-based approach. This is because the reward is a scalar projection of the, in general, n-dimensional state space that governs the dynamics of the task. This reduction in information makes the learning task much more challenging as the partial observability makes estimating the value and advantage functions more difficult. Exploration is also negatively impacted by the lower information feedback. If we have sparse rewards, the problem is even worse, as there can be large periods with no reward.

**Feasibility.** Despite these difficulties, it is still possible to train a reward-based policy for at least some simple environments. Let us consider a simple 1D Pointmass environment, where the reward is defined as a negative distance to the origin. In this environment, a reward-based agent can easily estimate its current position from the given reward if it correctly determines the sign by checking the previous action and reward changes. Even in $N$D Pointmass environments, we can design an analytical algorithm similar to a triangulation method that determines the location from a set of angle measurements. Therefore, it seems to be true that we can train a reward-based policy using off-the-shelf algorithms; it just takes a nearly exponential amount of time with respect to the difficulty of the task.

Figure 2 shows how the difficulty of learning the task scales as the number of dimensions for Pointmass increases. The maximum reward of policies with direct state access is impacted very little

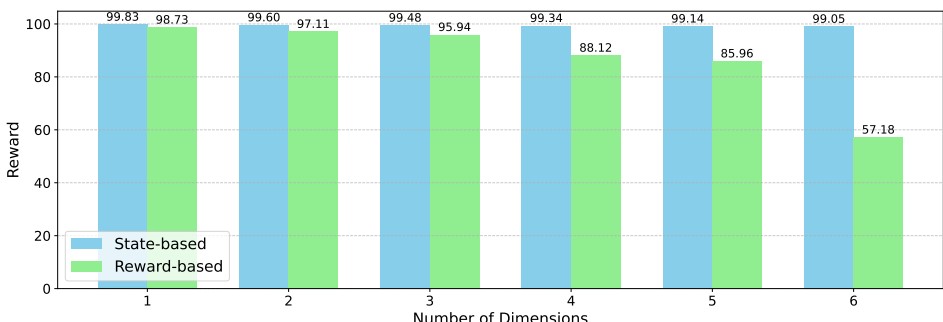

Figure 2: Effect of increasing state dimension on performance of state-based and reward-based policies. It shows that the performance of the reward-based policy decreases much faster than that of the state-based policy.

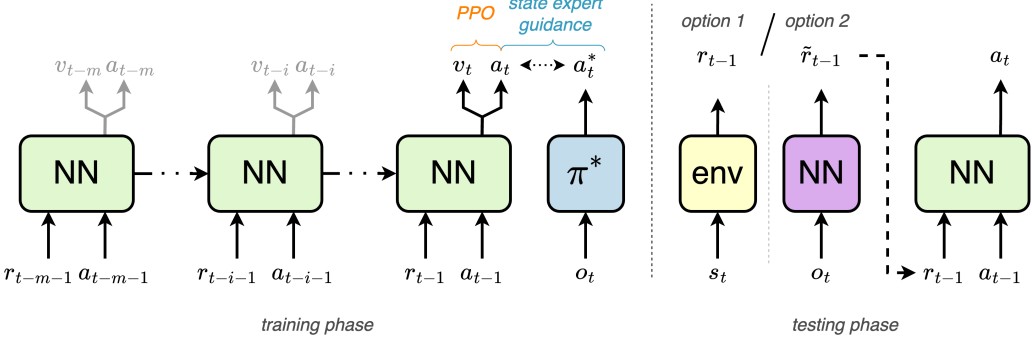

Figure 3: Architecture. The reward-based policy $\pi_R$ is trained using expert state-based guidance from $\pi^*$ and standard training in RL with PPO.

by the dimensionality of the problem, almost reaching the maximum possible reward of 100 for all scenarios, but the performance of the reward-based policy is heavily affected, only achieving $57\%$ of the maximum reward in the 6D Pointmass environment.

**Requirement for Dense Rewards.** Another significant consideration is the requirement for a dense reward function. Because the reward is being used as a signal for solving the MDP, it should be a dense function that gives a good value throughout the state space, while a sparse reward would be much worse in general and often impossible. Let us consider the Pointmass problem again. If the reward is sparse and only given at the goal, the agent will not receive any meaningful learning signal and will be unable to learn any useful behaviors.

**Differences in Optimal Behaviors.** Also important to note is the inherent difference in behavior between the observation-based and reward-based policies. Let us consider a 2D Pointmass environment. Given this very simple environment, an observation-based agent with direct access to the states $x$ and $y$ can easily learn the perfect actions to take to reach the goal directly. However, the reward-based agent will not be able to make an optimal action at the beginning because the only information that is available immediately from a single reward/action pair is the distance to the goal. The agent must take at minimum two information-gathering actions to localize itself from the history; then, it can take optimal actions based on the localization result. Given this suboptimal behavior for this very simple environment, it is fair to assume that learning will become much more challenging in more complicated environments.

### 3.3 PRACTICAL FRAMEWORK FOR LEARNING A REWARD-BASED POLICY

Despite the discussed difficulties, we found that two key features can effectively improve the training process of a reward-based policy:

1. Temporal history of the reward/action pairs.
2. Guidance of an expert policy, such as an observation-based agent.

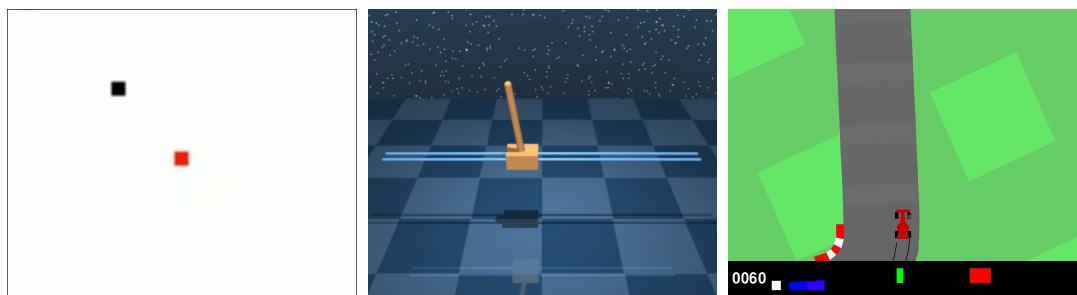

Figure 4: Tasks used for evaluation: Pointmass, Cartpole, and Car Racing.

To handle the temporal history, we first give the previous reward/action pair as inputs to a standard MLP for feature extraction. These extracted features are then passed to a recurrent network. We adopt an LSTM (Hochreiter & Schmidhuber, 1997) as a network architecture, but any recurrent network should perform similarly. The output of the LSTM is given as an input for the policy and value networks, which are trained with RL using Proximal Policy Optimization (PPO) (Schulman et al., 2017), using code modified from Huang et al. (2022). It is essential for control that some recurrent network is used for the inputs. In our experience, a single reward/action pair was not sufficient for effective control in most environments. On the other hand, the temporal history from the LSTM allows agents to estimate complicated contexts, particularly in more complex environments.

While a temporal history is necessary for control, we find that it is not sufficient for effective learning in practice. This degraded learning can be because a new highest reward record implies an unseen situation to the agent. To improve policy learning, we include an additional expert guidance loss along with the PPO loss from RL, where we train observation-based experts using a standard reinforcement learning algorithm. This loss is a supervised loss to regress the current policy actions to the actions of the expert policy. The full loss for training is $L_{PPO} + L_{guidance}$, where $L_{guidance} = (a_t - a_t^*)^2$.

A full schematic of our architecture is shown in Figure 3.

## 4 RESULTS

We evaluate The proposed reward-based policy on multiple tasks to answer the following questions:

1. Can you train a reward-based policy that only uses reward and action information instead of observations? How does a reward-based policy perform compared to a standard observation-based policy?

2. If we assume access to rewards, can a reward-based policy achieve zero-shot transfer to new problems with completely different observations?

3. If we must estimate reward information at inference time, how effectively can we transfer with a reward-based policy?

### 4.1 ENVIRONMENTS

We evaluate our work on three separate tasks shown in Figure 4. They are as follows:

1. **Pointmass:** 2D Pointmass environment with kinematic actions: $s_{t+1} = s_t + a_t$.

2. **Cartpole:** A Cartpole task from the DeepMind Control Suite (Tassa et al., 2018).

3. **Car Racing:** Gymnasium (Towers et al., 2024) Car Racing. Because this task's reward is a sparse reward that is only provided when crossing new pieces of the race track, we modified it to be continuous. Our modified reward follows the form of the rewards from the DM Control Suite: $r_{\text{modified}} = r_{\text{distance}} r_{\text{angle}}$, where $r_{\text{distance}} = \exp(-d)$, where $d$ is the distance to the next track waypoint, and $r_{\text{angle}} = \cos(\theta)$, where $\theta$ is the difference in track angle and the car's heading.

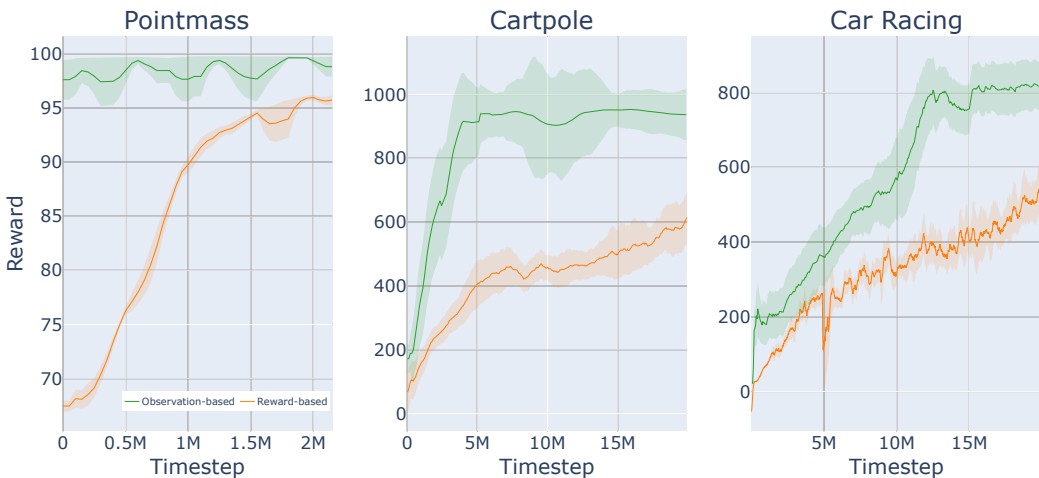

Figure 5: Learning curves for reward-based policies and observation-based policies. Observation-based policies serve as an upper bound on performance.

| Task | Original Observation | Shifted Observation |
|------|---------------------|--------------------|
| Pointmass | $97.06 \pm 1.36$ | $95.96 \pm 2.47$ |
| Cartpole | $758.74 \pm 62.79$ | $755.45 \pm 52.63$ |
| Car Racing | $775.37 \pm 225.27$ | $715.93 \pm 264.28$ |

Table 1: Zero-shot transfer for different observations via swapping color palettes. Mean value $\pm$ standard deviation is reported over 50 trials. Given the standard deviation, the behavior is functionally identical in both observation domains.

### 4.2 SUCCESSFUL TRAINING OF REWARD-BASED POLICIES

For all three environments, we are able to train successful reward-based policies. The learning curves from training across three random seeds are shown in Figure 5. We compare our reward-based policies to a standard observation-based policy as an upper bound on the performance: reward-based policies demonstrate 95%, 66%, and 70% performance of state-based policies in three environments, respectively. While they could not match the exact upper bound performance in the allotted training time, the learned reward-based policies demonstrate reasonable behaviors. For instance, a reward-based Pointmass agent can navigate to the goal with a few exploratory movements. In the Car Racing environment, a reward-based agent does not always follow the centerline of the track to get more reward signals and shows some sub-optimal turns. For qualitative comparisons, please refer to the accompanying video.

### 4.3 ZERO-SHOT TRANSFER VIA DIRECT REWARD ACCESS

Assuming we have access to the reward at test time, one intuitive application of our method is zero-shot transfer of a policy under observation shifts. If we take any of our environments and do a color palette swap, either randomly chosen or intentionally designed to be difficult, our method can transfer seamlessly. On the other hand, a standard observation-based policy would likely need to be retrained for each specific grouping of colors. Example color shifts for our environments are illustrated in Figure 6 and the effectiveness of our policy for the original and modified environments is demonstrated in Table 1.

Another application we tested is for transferring a Car Racing policy. However, instead of a relatively simple color palette swap, we consider a more dramatic observation shift, from the simple 2D visualization of the Gymnasium Car Racing environment to the photorealistic 3D render from Microsoft AirSim (Shah et al., 2017). These two environments are semantically similar and have the same action spaces and rewards. However, the dynamics are very different, which breaks our

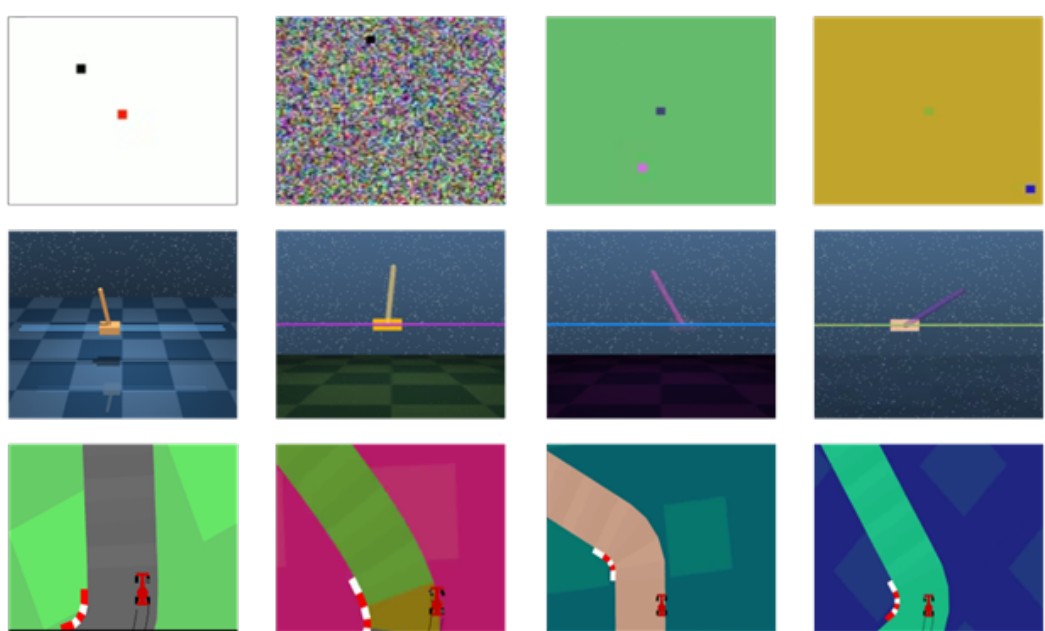

Figure 6: A reward-based policy can solve the task with any observation shifts in a zero-shot manner.

initial assumption in Section 3.1. Because our focus is not on dealing with the dynamics shift, we consider transfer with the simplified dynamics. Therefore, we match the transition function of 3D AirSim environment to that of 2D Car Racing environment by reimplementing its dynamics. Once the transitions are matched, a reward-based policy is successfully transferred to the 3D environment and demonstrates reasonable driving performance that travels for over 60 seconds with three turns. This was done as a proof of concept in a residential neighborhood in a grid pattern, but given the right configuration of waypoints, any trajectory seen in the 2D Car Racing training data should be achievable.

## 4.4 RAPID TRANSFER USING ESTIMATED REWARD

In some circumstances, we may not have access to the reward at inference time. We demonstrate that our method can still provide benefits in these scenarios. For the DMC Cartpole task, we examine the problem of transferring a policy when dealing with an image-based observation. Assuming access to a trained state-based policy and reward-based policy, we seek to learn an encoder that can map the image to the states and the rewards so that the policy can be directly reused in the target image domain. With each policy, we collect 100000 samples from the target image domain and use supervised learning to train an encoder to map to the original policy domain. For the given amount of data, we show that the reward can be effectively estimated and our reward-based policy can be deployed with a small performance decrease while the state encoder fails to accurately estimate the state and performs as effectively as a random policy. These results are shown in Table 2. A reward-based policy with the learned estimator still works well even when the reward is not available, which is much better than a state-based policy with a learned state estimator. This result demonstrates the effectiveness of using rewards for transfer, even if the performance of a reward-based policy in the source environment is suboptimal.

We hypothesize that our method works better as it is easier to estimate the 1 dimensional reward than it is the higher dimensional (five, in this case) state as it is less susceptible to noise and overfitting. This matches other works' observations on using low-dimensional latent spaces for control rather than the full state space. While we could perhaps get a similar result with a trained 1D latent space, we think our method is better as the reward is an inherent component of the MDP and requires no training. In addition, a reward is interpretable as we know what the reward is and it also offers direct, zero-shot transfer for environments with the same reward.

| Policy | Original Domain | Image Domain |
|---|---|---|
| State-based | $915.49 \pm 2.24$ | $189.41 \pm 12.54$ |
| Reward-based | $836.49 \pm 4.46$ | $735.83 \pm 65.48$ |
| Random | - | $192.78 \pm 36.91$ |

Table 2: Benefit of a reward-based policy even when the reward is not available at inference. Estimating an approximate reward from an image is easier than a full state, so the reward-based policy can be used directly from a small set of training data where the state-based policy fails. Means and standard deviations are reported over 100 trials with each policy.

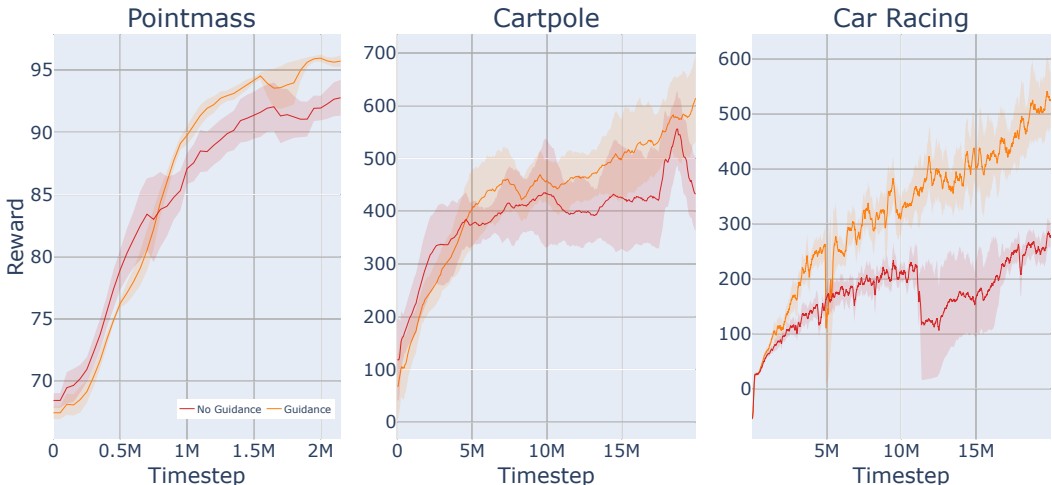

Figure 7: Ablation for expert guidance. Additional guidance term offers slight benefit for Pointmass and Cartpole but is very beneficial for Car Racing.

### 4.5 ABLATION ON EXPERT GUIDANCE

To validate the design choice of including expert state-based guidance during training, we compare reward-based policies trained with and without guidance in Figure 7. This shows that for the Pointmass and Cartpole, extra guidance gives a small performance boost, while for the Car Racing task, it achieves much higher rewards. This is possibly caused by the relative difficulty of the tasks, as in both the Pointmass and Cartpole tasks failure to effectively act just reduces the maximum reward but the agent can try again, whereas for the Car Racing task, failure to maintain position on the track terminates the episode.

## 5 CONCLUSION

This paper presents a novel approach to solving MDPs using policies with only rewards and actions as inputs. We examine the feasibility and practical methods for learning such reward-based policies, requiring temporal context provided by a recurrent network architecture and expert guidance provided by an observation-based expert policy. We show the benefits of such policies for zero-shot transfer among environments with the same dynamics and different observations, assuming direct access to the reward at inference time. We also demonstrate the potential usefulness of our method even when we do not have access to the reward and must estimate it.

There are several directions for future work. In this work, we assume identical transition functions between the source and target environments to focus on observation shifts. However, a reward-based policy could potentially handle changes in dynamics by augmenting its training process with common transfer techniques such as domain randomization or system identification. Additionally, we evaluate the concept of a reward-based policy in relatively simple environments. It would be

interesting to explore their potential in more realistic domains, such as real-world robotic control or autonomous navigation.

**Reproducibility Statement**

- We provide hyperparameters for PPO in the Appendix A.1.
- We use standard RL benchmarks from Tassa et al. (2018) and Towers et al. (2024).

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

# A APPENDIX

## A.1 PPO HYPERPARAMETERS

| Parameter | Value |
|---|---|
| Learning rate | $3 \times 10^{-4}$ |
| Steps per update | 10000 |
| Batch size | 10000 |
| $\gamma$ discount | 0.99 |
| GAE $\lambda$ | 0.95 |
| Clip range | 0.2 |
| Gradient clipping threshold | 0.5 |
| Update epochs | 50 |
| $v_f$ coefficient | 0.5 |

