# OpenReview forum: "Reward as Observation: Learning Reward-based Policies for Rapid Adaptation"
_ICLR.cc/2025/Conference — Submitted to ICLR 2025_

### Official Review · Reviewer_tLqz · 2024-10-28

**Soundness:** 2
**Presentation:** 2
**Contribution:** 2
**Rating:** 3
**Confidence:** 4

**Summary:**

This paper tackles the challenge of enabling intelligent agents to generalize to unseen environments by focusing on changes solely in observation spaces. The authors introduce a novel approach that trains policies using only the history of rewards and actions, excluding direct observations. They hypothesize that dense reward signals can facilitate zero-shot transfer across diverse environments. The proposed framework leverages temporal histories of reward-action pairs and expert guidance to train reward-based policies. Experiments in Pointmass, Cartpole, and Car Racing demonstrate that these policies achieve 60% to 90% of the performance of standard observation-based policies while maintaining transferability to environments with significant observation shifts.

**Strengths:**

- **Novel Approach:** The paper presents an innovative method for training policies using only reward and action histories, offering a fresh perspective in reinforcement learning.

- **Intuitive Explanation:** The authors clearly and convincingly explain why reward-based policy learning can be effective, especially in navigation tasks. For example, learning a goal-conditioned policy remains feasible by only incorporating the reward signal.

**Weaknesses:**

- **Limited Real-World Applicability:** Training without direct observations restricts the method’s applicability, as real-world scenarios often involve more complex changes beyond observation shifts. This limitation raises concerns about the generalizability of reward-conditioned policies in diverse settings. What are the possible scenarios in the real world where only observation changes within the MDP?
- **Scalability Issues:** While the reward-based learning approach shows promise in simpler environments like Cartpole, its effectiveness in more complex tasks such as locomotion, manipulation, or humanoid control remains uncertain. Can reward-based policy handle complex locomotion or manipulation tasks? Since this work focuses on understanding transfer benefits, there is concern that it might struggle to learn necessary behaviors before transfer.

**Questions:**

- **Experimental Clarity**: The illustrative example referenced in Figure 1 is missing from the paper (there is no real-road transfer), which may confuse readers. The authors should replace it with an existing figure from their experiments.
- **Generalization to Varied MDPs**: Beyond identical observation shifts, how does the reward-conditioned policy perform under broader MDP changes, such as variations in transition dynamics or reward structures?

---

### Official Review · Reviewer_4opR · 2024-11-01

**Soundness:** 1
**Presentation:** 3
**Contribution:** 1
**Rating:** 3
**Confidence:** 4

**Summary:**

This paper explores a reward-only approach to RL, where policies are trained using rewards and actions rather than observational data, enabling potential zero-shot transfer across environments with different visual representations. The authors propose a method using LSTM-based temporal history and expert-guided behavior cloning to learn reward-based policies in simple environments. They demonstrate that reward-only policies perform reasonably well compared to observation-based ones, especially in tasks with consistent dynamics. The method is evaluated in popular tasks like Pointmass, Cartpole, and Car Racing, with 2D-to-3D transfer shown in a reconfigured AirSim environment.

**Strengths:**

**S1. Thorough Challenge Analysis**: The authors provide a useful in-depth examination of the difficulties inherent to reward-based policies, such as poor observability, difficulty in estimating value functions, and limited exploration. This highlights the limitations of using a scalar reward signal rather than high-dimensional state information. They emphasize the necessity of dense rewards, showing how the method struggles as dimensionality increases.

**S2. Expert Guidance Ablation.** An ablation study on the behavior cloning loss component shows how important it is, especially in the more complex task like Car Racing.

**S3. Simplicity and Comprehensibility.** The method is straightforward, with minimal components and no additional tunable hyperparameters, making it easy to implement and understand.

**S4. Robustness to Observation Quality**: Once trained, the model is independent of observation quality, allowing it to perform well even with noisy or degraded visual observations.

**S5. Effective Reward Estimation for Transfer.** In Section 4.4, the authors demonstrate the transfer capabilities by estimating rewards directly from images, allowing the reward-based policy to perform well even without access to the true reward at inference. They further show that transferring from a 1D reward signal is easier and more reliable than transferring from high-dimensional observations, requiring minimal data and proving more resilient than state estimation.

**Weaknesses:**

**W1. Numerous Assumptions.** The method makes many assumptions for successful implementation, limiting its adaptability and real-world applicability.

1. **Dense and Accurate Rewards**. The authors state that rewards must be *sufficiently dense* and *give a good value throughout the state space*, yet there is no analysis of how sparse they need to be. This lack of clarity makes it uncertain how sparse or dense rewards can be before training fails. In practice, it likely means that transitions without an informative reward signal contribute minimally to training, making the approach highly sample-inefficient. In realistic settings, feedback is often imperfect or delayed. For example, in robotics, sensors might provide inaccurate readings due to interference or hardware limitations, leading to noisy or inconsistent rewards. Similarly, in environments where rewards are human-generated (like feedback in recommendation systems), subjective or inconsistent responses can introduce noise.
2. **Domain Compatibility**. The source and target environments should share the same transition dynamics, action and observation spaces, and reward structure. This overlooks many practical cases, where slight discrepancies in dynamics or observation structures are the norm. The authors don’t discuss where these conditions might realistically apply, leaving practical feasibility unexplored.
3. **Expert Guidance**. The ablation study shows that expert guidance significantly boosts performance, particularly on complex tasks. However, this dependence undermines the claimed benefits of a “reward-only” approach, as it reintroduces standard RL processes the authors aim to bypass. Moreover, training an observation-based expert policy adds a computational overhead, and since training is done online, inference also needs to be run on the expert model.

**W2. Loss of Spatial Context.** By relying solely on rewards and actions, the method lacks spatial awareness, which is critical in tasks requiring an understanding of position or orientation. For example, in navigation tasks like maze-solving, an agent without spatial context will struggle to differentiate between distinct but similarly rewarding areas, such as identical-looking corridors or dead ends. In manipulation tasks, the lack of positional feedback leads to incorrect actions, like reaching for objects without adjusting for their relative location. Without such spatial cues, the method is limited to simple tasks where positioning doesn’t play a role. The agent’s ability to keep the car centered on the track in the Car Racing environment hinges on the overly-engineered reward function that promotes this behavior. Relying on such finely-tuned rewards is impractical in real-world applications, where crafting reward functions to this level of precision is rarely feasible.

**W3. Lack of methodological novelty**. The approach lacks innovation, relying on a simple combination of an LSTM and behavior cloning to train PPO. This simplicity offers little advancement over existing techniques and contributes minimally to the field, as both components are well-established in RL. Moreover, the explanation of the method is scarce. Neither the text nor the caption of Figure 3 adequately explains the diagram, making it difficult to interpret. For example, it’s not immediately evident that the ‘options’ in the testing phase of Figure 3 depict the modes where the environment provides the reward and where the reward is learned.

**W4. Weak Experimental Evaluation.**
1. The experiments are limited to simple environments (Pointmass, Cartpole, and Car Racing), with the authors suggesting that more complex environments with higher state dimensions or sparse rewards are "impossible" for this method. This raises questions about practical utility—if the method can't handle harder, more realistic tasks, its applicability remains unclear.
2. The authors only compare their reward-based policy to a regular observation-based policy, which is supposed to serve as an upper bound. However, they include no comparisons with other established techniques whatsoever, offering no context on how their approach measures up to alternative methods in this setting.
3. While evaluation is done over 50 trials, the number of seeds used for training isn’t specified. Without this, it’s hard to assess the robustness of the method.
4. The authors state that *it is essential that some recurrent network is used*, and that that *a single reward/action pair is not sufficient*. However, they provide no analysis or ablation study to clarify the LSTM’s specific contribution. Alternative approaches for maintaining temporal history, like stacking previous rewards and actions, are not explored, leaving it unclear whether the LSTM is genuinely necessary or if simpler methods could achieve comparable results.

**W5. Transfer Limitations**. The authors attempt 2D-to-3D transfer by using the Car Racing and AirSim environments, as shown in Figure 1, considering a scenario where changes only occur in pixel-based observations. This leads them to circumvent the realistic challenges of transfer by manually reimplementing the kinematics to match across environments. This undermines the notion of true zero-shot transfer, as manual alignment is rarely viable in real-world applications and reveals the method’s limited applicability. Furthermore, no quantitative results are provided for the 3D transfer; it’s merely claimed that the policy shows “reasonable driving performance,” leaving the success of this transfer largely unsubstantiated. The authors’ statement that “our focus is not on dealing with the dynamics shift” dismisses the real complexities of 2D-to-3D transfer, where adapting to dynamic differences is unavoidable. This approach fails to demonstrate genuine transfer.

**Questions:**

Q1. How does the reward-based policy compare to model-based methods that learn the dynamics of the environments?

Q2. What’s the broader motivation for this approach, given that it currently only works in very basic environments and requires nearly exponential time as task complexity increases, if it even works at all?

Q3. Based on what does the agent make the first move at the start of an episode before any rewards have been granted?

Q4. Why did the authors choose the color palette swap to introduce the observational shift? Perhaps it should be explained that any change in the observation makes no difference because it is not regarded as part of the input to the model.

Q5. Why not also include an ablation study for the RNN part?

Q6. What do “sufficiently dense” rewards entail for effective training?

Q7. How does the method handle noisy or inconsistent rewards, particularly in real-world applications where feedback may be delayed or imperfect?

Q8. What practical scenarios do you envision where the source and target environments would have identical dynamics, actions, observations, and reward functions?

Q9. How does reliance on expert guidance align with the goal of a “reward-only” approach?

Q10. How would the method handle tasks requiring spatial awareness, such as navigation or manipulation tasks?

Q11. How does the method compare to other established RL techniques?

Q12. What are the quantitative results of the transfer experiments?

---

### Official Review · Reviewer_s23V · 2024-11-04

**Soundness:** 2
**Presentation:** 4
**Contribution:** 1
**Rating:** 1
**Confidence:** 5

**Summary:**

This paper introduces a method called reward-based policy. It makes the assumption that rewards are observable in a reinforcement learning problem, and attempts to learn a policy that outputs actions based only on previous rewards and actions, and no states or observations. The hope is that such a policy will be transferable between environments where dynamics (and reward functions) are the same.

**Strengths:**

The paper is very well-written and very easy to follow and understand.

**Weaknesses:**

I think the major problem with the paper is that the studied setting is very unrealistic. The paper provides some discussions, which also acknowledge how difficult it is for their assumptions to be satisfied. At the end, it is still difficult to think of any application where the proposed approach is going to be useful.

Specifically, I am not worried about the assumption that the reward is observable. But what are some real-world problems / environments where an RL agent can perform reasonably well just by seeing the previous actions and rewards? As the paper acknowledges, this is exactly like trying to play a game by looking only at the score and no other parts of the screen. Other than environments where it is possible to memorize the solution, it seems to me that these conditions are extremely hard to satisfy. And this is not even the only assumption. In addition to these, there should also be some transfer concern where the robot's transition and reward functions stay the same but the observations change. I don't see any useful application.

A somewhat promising part of the paper is the section where it tries to estimate rewards from observations. However, that section requires access to a state-based policy that solves the task. Again, this is very unrealistic. If the problem is a POMDP, how would one have access to a state-based policy? The states are not even known to the agent.

Finally, the main promise of the paper is not very interesting. When the learned policy does not depend on the observations, of course the observation function can be changed in any arbitrary way. Under that setting, that function is completely irrelevant. So the experiments are completely unsurprising and obvious.

Below are my other comments that are more minor:
- et al. is plural, so those citations should be thought as "they" instead of "it". There are some grammatical errors about this.
- The POMDP definition is missing the function that maps the states to observations.
- The so called reward-based policy is defined as $R\times A \to A$ but my understanding is that a full history of rewards and actions are inputted, not just the most recent ones. In that case, this definition of the policy function is incorrect.
- Incorrect capitalization in line 248.
- Again, the second hypothesis/question (line 253) is validated by the definition of reward-based policy. Why is it even a hypothesis?

**Questions:**

What are some real applications this method will be useful?

---

### Official Review · Reviewer_CkYD · 2024-11-05

**Soundness:** 1
**Presentation:** 3
**Contribution:** 1
**Rating:** 1
**Confidence:** 4

**Summary:**

The authors propose a method to learn transferable policies by treating rewards as the observation space on which a learned policy operates. Specifically, they learn behavior policies that utilize as input, a history of observations and actions and generate future actions. These inputs are aggregated using LSTMs which generate actions. This model is trained using PPO online along with an offline behavior cloning loss on a demonstration dataset. The method is evaluated on 3 simple environments - Cartpole, pointmass and Racing Car.

**Strengths:**

1. The paper is well written and easy to understand
2. The method is principled and intuitive.
3. The authors perform ablations and provide statistically significant results.

**Weaknesses:**

1. **Lack of a justifiable motivation:** It's unclear why one would use only rewards as the observation space in practice. For any complex dynamical system, having a policy dependent on an observation is required. Theoretically, this would only work in stationary bandit-like settings, where a single action optimal action exists from the ones available and a history of past actions and rewards would suffice to take the right action. However, in all other “RL” settings where an action taken changes the state of the world, i.e., environments with a transition function, this method would break. Also, it’s hard to motivate this method from a practical perspective - in most real applications, one would utilize all possible information available to learn a behavior. In short, the authors claim they study a much more complex problem - that of reinforcement learning, while applying a primitive set of assumptions - those of a multiarm bandit. I don’t believe this method would work under the advertised conditions. Additionally, I question the claims of generalization in the paper - but of course, the method would generalize across visual observation perturbations. It is not conditioned on these perturbed inputs.
2. **Unclear why this works:** I suspect the method is able to solve tasks due to the demonstrations available to it - not due to the online PPO. This is clear in the ablations also - for tasks that actually have a transition function like Car Racing, the demonstrations yield appreciable improvements over not using them.
3. **Lack of Novelty:** It’s unclear to me whether this paper adds to existing knowledge in any way. Multiarmed Bandits are well studied and memory based agents are well studied also. The paper currently does not present any new theoretical insights, nor does it show massively scaled experimentation.

**Questions:**

1. How big was the demonstration dataset used to train the method?
2. How were losses balanced? Was RL performed in parallel to supervised learning?

---

### Meta-Review · Area_Chair_oTRx · 2024-12-22

**Metareview:**

The manuscript appears unready for publication, with several major concerns raised by reviewers.

**Additional Comments On Reviewer Discussion:**

The authors declined to comment on the reviewers' concerns.

---

### Decision · Program_Chairs · 2025-01-22

Reject